# The footprint of colour in EEG signal

## Abstract

Our perception of the world is inherently colourful, and colour provides well-documented benefits for vision: it helps us see things quicker and remember them better. We hypothesised that colour is not only central to perception but also a rich, decodable source of information in electroencephalography (EEG) signals recorded non-invasively from the scalp. While previous work has shown that brain activity carries colour information for simple, uniform stimuli, it remains unclear whether this extends to natural, complex images with no explicit colour cueing. To investigate this, we analysed the THINGS EEG dataset, which contains 64-channel recordings from participants viewing 1,800 distinct objects (16,740 images) presented for 100 ms each, yielding over 82,000 trials. We established a perceptual colour ground truth through a psychophysical experiment in which participants viewed each image for 100 ms and selected the perceived colours from a 13-option palette. An artificial neural network trained to predict these scene-level colour distributions directly from EEG signals showed that colour information was robustly decodable (average F-score of 0.5). We further examined the effect of colour features on object decoding. Using a contrastive learning framework, we modelled colour–object perception with the Segment Anything Model (SAM), in which all pixels within a segment were replaced with their average colour, followed by standard feature extraction using CLIP vision encoders. We trained an EEG encoder, CUBE (ColoUr and oBjEct decoding), to align features in both object and colour spaces. Across EEG and MEG datasets in a 200-class recognition task, incorporating colour improved decoding accuracy by approximately 5%. Together, these findings demonstrate that EEG signals recorded during natural vision carry substantial colour information that interacts with object perception. Modelling this interaction enhances the power of neural decoding.

## 1 How strong is the colour signal in neuroimaging?

Our visual system makes sense of a scene with remarkable speed. In just a fleeting glance, as brief as 13 ms, we can attach a simple description such as "green tree" to what we have seen (Potter et al., 2014). This raises a critical question: what neural representations emerge within such a brief window, and to what extent can they be captured in neuroimaging signals? Here we turn our attention to colour, an effortless and ever-present aspect of vision. Colour not only shapes how we perceive objects (Tanaka et al., 2001; Bramão et al., 2011), but also enhances memorability (Gegenfurtner & Rieger, 2000; Wichmann et al., 2002) and speeds up recognition (Møller & Hurlbert, 1996; Rosenthal et al., 2018).

Colour decoding from neuroimaging has a long history (Regan, 1970; Paulus et al., 1984). Brain activity carries information about chromaticity, luminance, and saturation (Sutterer et al., 2021; Hermann et al., 2022; Pennock et al., 2023; Rozman et al., 2024), the hue circle (Hajonides et al.,

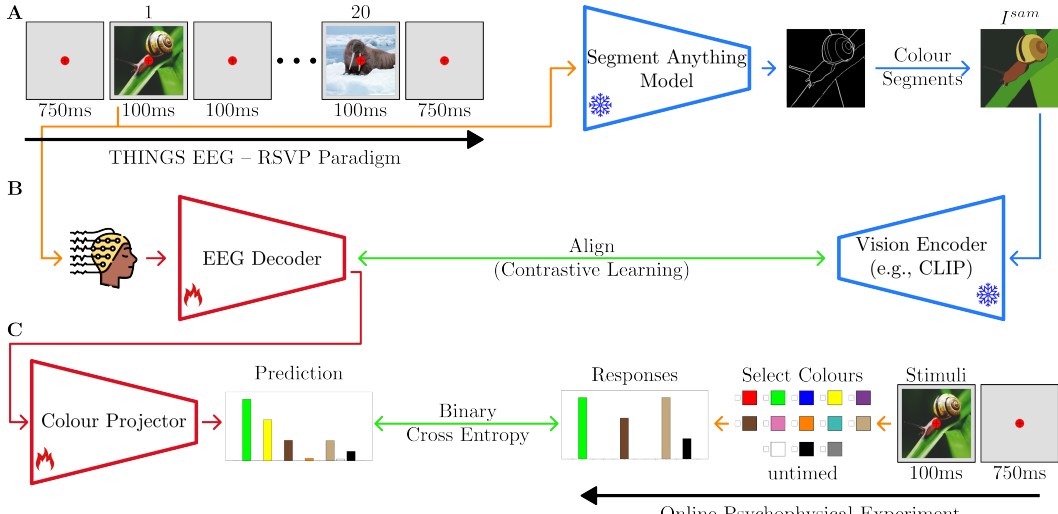

Figure 1: The overview of ColoUr and oBjEct decoding—CUBE. A: The RSVP paradigm used to collect the THINGS EEG dataset (Gifford et al., 2022). B: An EEG decoder aligns brain activity with features from a pretrained vision encoder applied to colour-segmented images. C: A linear projection layer maps the aligned representation onto the behavioural colour responses.

2021), the geometry of colour space (Rosenthal et al., 2021), and even unique hues (Chauhan et al., 2023). Fewer studies, however, have examined how colour interacts with object processing. One study suggests that while both shape and colour can be decoded as early as 60–70 ms after stimulus onset, shape–colour congruency emerges later, around 200 ms (Teichmann et al., 2020). These findings are valuable but mostly derive from simplified displays of uniform coloured patches on plain backgrounds. It remains unclear whether colour can be reliably decoded from brain signals when viewing rich, natural scenes—this is the challenge addressed in the present study.

Artificial intelligence and large datasets have recently accelerated progress in decoding. The NSD dataset (Allen et al., 2022), for example, provides large-scale fMRI data covering about 10,000 natural images. Similarly, the THINGS EEG (Gifford et al., 2022) and MEG (Hebart et al., 2023) datasets offer recordings of comparable scale, enabling new opportunities to investigate how natural images are represented in the brain. Alongside these resources, contrastive learning (Radford et al., 2021) has emerged as a powerful tool for decoding. It has already shown strong performance across modalities, from speech recognition (Défossez et al., 2023) to visual object recognition in fMRI (Scotti et al., 2024), EEG (Song et al., 2024), and MEG (Wu et al., 2025).

## 1.1 CUBE (ColoUr and oBjEct decoding)

We adopted the same general framework of large datasets and contrastive learning to investigate how colour is represented in brain signals during natural image viewing. Our focus here is on EEG, which, despite its low spatial resolution, offers high temporal resolution, affordability, portability, and the possibility of real-time decoding (Benchetrit et al., 2023; Robinson et al., 2023). Instead of collecting a new dataset, we created colour annotations for the THINGS EEG dataset (Gifford et al., 2022) through a large-scale psychophysical experiment designed to mimic the conditions of the original recordings. Participants viewed an image for 100 ms and then selected all perceived colours from a palette of 13 options (see Figure 1, panel C).

We trained an EEG decoder, implemented as a simple artificial neural network (ANN) with two linear layers and a residual connection, to align EEG representations with visual features extracted from pretrained CLIP networks. To better capture perceptual colour structure, which operates at the object rather than pixel level (Gegenfurtner, 2025), we processed each image $I^{org}$ using the Segment Anything Model (SAM) (Kirillov et al., 2023). For each segmented region, we averaged pixel colours to create uniformly colour-segmented images $I^{sam}$ (see Figure 1, panel B), providing a closer approximation to colour perception at a glance.

To evaluate colour decoding from EEG signals, we added a linear projection layer atop the CLIP-aligned features to output a 13-dimensional vector matching the behavioural colour palette. Performance was measured with the F-score for this multi-class task. Results show reliable decoding, with an average F-score of 0.50 across participants—well above chance (0.17). This constitutes our first contribution: demonstrating colour decoding from EEG during natural image viewing with 100 ms exposure. Notably, the noise ceiling in this rapid serial visual presentation (RSVP) paradigm is 0.64, indicating the decoder approaches average human agreement.

Encouraged by these results, we hypothesised that incorporating colour features into the contrastive alignment framework could improve object decoding. Colour and object perception are closely intertwined, both behaviourally (Bramão et al., 2011; Gegenfurtner, 2025) and neurally (Rosenthal et al., 2018; Tanaka et al., 2001). To model this, we aligned the EEG decoder simultaneously to CLIP features from the original images $I^{org}$ and colour-segmented images $I^{sam}$ (Figure 2), which capture object–colour associations more directly, particularly under the brief 100 ms exposure.

We term this framework CUBE (ColoUr and oBjEct decoding), as it explicitly leverages the interaction between colour and object representations in the brain. Experimentally, CUBE improves state-of-the-art object recognition decoding by a consistent 5% across all participants and in both EEG and MEG. This underscores the importance of colour–object interactions in neuroimaging decoding and points to a strongly shared representational space for colour and object in the brain.

## 2   Method

We primarily focused on the THINGS EEG dataset and, secondarily, on the MEG dataset, both derived from a subset of the THINGS collection (Hebart et al., 2019), a high-quality set comprising 1,854 diverse object concepts. We generated colour annotations for the images through an online psychophysical experiment and employed a contrastive learning framework to train our networks.

### 2.1   Neuroimaging datasets

THINGS EEG (Gifford et al., 2022): Recordings were collected from 10 participants using an RSVP paradigm (Intraub, 1981), where each image was shown for 100 ms, followed by a 100 ms blank (Figure 1, Panel A). EEG was recorded with a 64-channel cap. The training set included 1,654 concepts (10 images per concept, 4 repetitions per image), and the test set 200 unseen concepts (1 image per concept, 80 repetitions per image). Preprocessing followed the original paper: signals were epoched 0–1000 ms post-stimulus, downsampled to 250 Hz, and reduced to 17 occipito-parietal channels most relevant to vision[1]. To improve signal-to-noise ratio, repetitions of the same image were averaged, yielding 16,540 training samples and 200 test samples per participant.

THINGS MEG (Hebart et al., 2023): Recordings from 4 participants with 271 channels, each image presented for 500 ms followed by a 1000 ± 200 ms interval. The training set included 1,854 concepts (12 images per concept, 1 repetition each), and the test set comprised 200 concepts (1

---

[1]P7, P5, P3, P1, Pz, P2, P4, P6, P8, PO7, PO3, POz, PO4, PO8, O1, Oz, O2

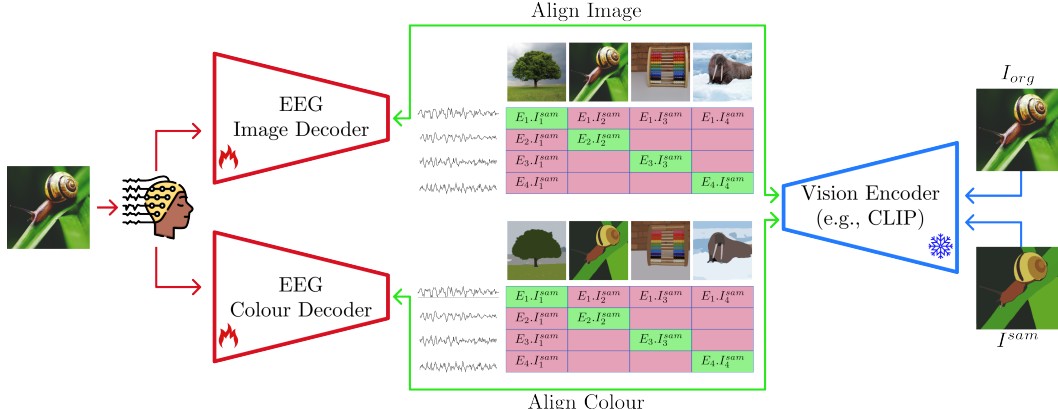

Figure 2: CUBE: Incorporating colour features into object decoding. Two EEG decoders are trained to align with (1) the original RGB images viewed by participants and (2) colour-segmented versions of the same images. Colour alignment follows contrastive learning (Radford et al., 2021), while image alignment employs the Uncertainty-aware Blur Prior algorithm (Wu et al., 2025).

image per concept, 12 repetitions each). Test concepts were excluded from training for zero-shot EEG evaluation. Preprocessing followed the same pipeline as EEG, with MEG signals downsampled to 200 Hz.

## 2.2 Psychophysical experiment on colour perception under brief exposure

When an image is shown for only 100 ms, a key question is how much of the scene can be understood and accessed consciously (Keysers et al., 2001). To approximate participants' colour perception in the EEG experiment, we conducted a psychophysical study with a similar presentation rate. A blank screen with a central fixation cross was shown for 750 ms, followed by the image for 100 ms. Participants, instructed via written guidelines, selected all perceived colours from a thirteen-colour palette (Figure 1, Panel C).

To annotate the full THINGS EEG dataset (16,740 images), the experiment was run online via Prolific (Palan & Schitter, 2018). Each participant saw 450 images. For training images, participants had 5 seconds per trial due to the large volume; for test images, no time limit was imposed to capture all perceived colours. In total, 133 participants from diverse cultural and linguistic backgrounds participated and were compensated monetarily.

Subjectively, this fast-paced paradigm shows that participants tend to recall only a few foreground objects and their colours. Strong colour–object associations emerge, while background colours are poorly remembered unless covering large uniform regions. This is reflected in the responses: participants selected on average 2.1 colours per image. Beyond the THINGS EEG colour annotations, this large-scale experiment provides a valuable dataset for studying colour perception under brief exposures in ecologically relevant settings, which is released for the community.

## 2.3 Visual neural decoding

To decode colour and object from neuronal data, we adopted a contrastive learning paradigm (Radford et al., 2021), which has been widely used in neuroimaging research (Song et al., 2024; Li et al., 2024; Scotti et al., 2024). In this framework, an EEG decoder is trained to align its output

representations with those of a pretrained vision encoder, often a variant of CLIP (Figure 2). Similar strategies have also been applied with other modalities, such as language encoders (Akbarinia, 2024) and depth encoders (Zhang et al., 2025).

Formally, let $\mathbf{z}_i^{EEG}$ denote the feature representation predicted by the EEG decoder for sample $i$, and $\mathbf{z}_i^{IMG}$ the corresponding representation from the vision encoder. The goal of contrastive learning is to maximise similarity between matching pairs $(\mathbf{z}_i^{EEG}, \mathbf{z}_i^{IMG})$ while minimising similarity with all non-matching pairs in the batch. This is achieved with a symmetric cross-entropy objective:

$$
\mathcal{L}_{\text{CLIP}} = -\frac{1}{N} \sum_{i=1}^{N} \left[ \log \frac{\exp\left(\text{sim}(\mathbf{z}_i^{EEG}, \mathbf{z}_i^{IMG})/\tau\right)}{\sum_{j=1}^{N} \exp\left(\text{sim}(\mathbf{z}_i^{EEG}, \mathbf{z}_j^{IMG})/\tau\right)} + \log \frac{\exp\left(\text{sim}(\mathbf{z}_i^{IMG}, \mathbf{z}_i^{EEG})/\tau\right)}{\sum_{j=1}^{N} \exp\left(\text{sim}(\mathbf{z}_i^{IMG}, \mathbf{z}_j^{EEG})/\tau\right)} \right],
$$
(1)

where $N$ is the batch size, $\tau$ is a learnable temperature parameter, and $\text{sim}(\cdot, \cdot)$ denotes the cosine similarity. This loss encourages the EEG and image representations of the same stimulus to be close in the embedding space, while separating them from mismatched pairs.

One persistent challenge in neuroimaging applications is dataset size: current datasets are relatively small for deep learning, leading to overfitting during training and poor generalisation at test time. A recently proposed technique, the Uncertainty-aware Blur Prior (Wu et al., 2025), mitigates this by introducing a foveated blur to the original images $I^{org}$, simulating how participants perceive stimuli. By suppressing high-frequency details, this strategy reduces one of the main drivers of overfitting. We adopt this approach in our training framework, processing $I^{org}$ with the foveation blur described in Wu et al. (2025).

Building on this idea, we propose a colour-aware contrastive learning framework, in which the decoder is additionally aligned with CLIP features extracted from colour-segmented images, denoted as $I^{sam}$. These images are derived from the original input $I^{org}$ using SAM-1 (Kirillov et al., 2023) with default global segmentation parameters, except for an increased resolution of 64 points per side and a stability score threshold of 0.92. We hypothesise that colour-segmented images more closely resemble participants' perceived colours and their object associations during brief exposures (100 ms). Consequently, introducing a contrastive loss term $\mathcal{L}_{\text{CLIP}}$ between $\mathbf{z}_i^{EEG}$ and $\mathbf{z}_i^{SAM}$ is expected to boost object decoding.

To evaluate this, we conducted pilot experiments in which participants viewed $I^{org}$ for 100 ms, followed by a 750 ms grey screen. After this interval, $I^{sam}$ was presented either alone or alongside a greyscale version of $I^{org}$. Participants were instructed to click on pixels whose colour values were inconsistent with the original scene viewed for 100 ms. Only a small number of mismatches were reported, suggesting that colour-segmented images provide a close approximation of perceived colours, objects, and their associations under such brief viewing conditions.

The EEG Decoder in CUBE follows Wu et al. (2025): two linear layers with GELU activation and a residual connection. The Colour Projector comprises two linear layers with ReLU, mapping CLIP-aligned features to a thirteen-colour palette. In all experiments, OpenCLIP (Cherti et al., 2023) was used to extract features from different vision encoders. Networks were trained for 50 epochs with batch size 1024 using the AdamW optimiser (Loshchilov & Hutter, 2017), with learning rate $1 \times 10^{-4}$ and weight decay $1 \times 10^{-4}$. Other configurations followed Wu et al. (2025).

For all experiments, two types of networks were trained. Intra-participant networks were trained and evaluated on the EEG data of the same participant, whereas inter-participant networks were trained on all participants except one, which was held out for testing.

## 3 Colour decoding

Colour decoding is inherently a multi-class task, as multiple colours may co-occur within a single image. Unlike object recognition, colour perception shows striking individual differences, particularly under brief viewing (Mollon et al., 2017; Bosten, 2022). For example, one participant may label wood as brownish, whereas another may choose a more beige shade (Lafer-Sousa et al., 2015). To accommodate this multi-class structure and the variability across observers, we quantified agreement between two human responses using the F-score:

$$F = \frac{TP}{2TP + FP + FN}, \tag{2}$$

where $TP$, $FP$, and $FN$ denote true positives, false positives, and false negatives, respectively. We chose the F-score over the closely related Jaccard index (Jaccard, 1901), which is also used for set comparison, because the F-score assigns greater weight to $TP$s. This emphasis is better suited to colour fidelity, as it highlights dominant colours selected by participants. We directly compared behavioural and neural data by applying the same metric to quantify agreement between EEG-decoded colours and the average human responses. Because neither the human averages nor the model predictions are binary, both were thresholded. All reported F-scores use a threshold of $\frac{1}{3}$, based on the rationale that at least one-third of participants agreed on a colour for a given image.

The results of colour decoding on the THINGS EEG dataset are shown in Figure 3. Overall, the CUBE model achieves an F-score above 0.50, approaching the noise ceiling (0.64)—the average agreement among participants—and well above chance (0.17), estimated over 10,000 iterations using two randomly selected colours per trial to match typical participant responses. Performance also exceeds a baseline of 0.23, computed similarly but sampling colours from the training-set distribution. Inter-participant models reach lower F-scores (0.33) yet still far exceed chance, indicating a shared representation of colour across individuals (Gegenfurtner, 2003).

Excluding object alignment from training results in colour decoding with an F-score of 0.46, significantly exceeding chance levels. This shows that the network can extract meaningful colour information directly from EEG signals without relying on any additional source of information. Nevertheless, for most participants, colour decoding improves significantly (Student's t-test) when object alignment is included, consistent with evidence that object and contextual features influence colour perception (Witzel & Gegenfurtner, 2018; Tanaka & Presnell, 1999; Gegenfurtner, 2025). Although the reported results use the CoCa-ViT-L-14 architecture (Yu et al., 2022), no significant differences were found when using alternative architectures.

The F-scores vary by 7% between the best and worst participants (53% for participant 06 and 46% for participant 04), yet the distributions across the test set appear qualitatively similar. This consistency likely reflects the fact that the colour ground truth is based on average human responses rather than each participant's individual colour perception during the EEG experiment (Bosten, 2022). Consequently, higher decoding accuracy might be achievable if the behavioural ground truth corresponded to the neural data of the same individual.

The examples in Figure S1 show that neurally decoded colours often remain plausible even in trials with low F-scores. For instance, the Flax Seed image is behaviourally labelled brown–beige, while the decoded hues are neighbouring orange–red, indicating only a small mismatch. Similar swaps between red and orange appear for the Omelette and Fruit images—a known challenge even for computer vision models (Parraga & Akbarinia, 2020). In other cases, such as the Elephant image, mechanisms like colour constancy (Akbarinia & Parraga, 2017; Gil Rodríguez et al., 2024) or memory (Hansen et al., 2006) may drive participants to report grey despite physically yellowish–beige pixels, reflecting processes requiring longer integration than the 100 ms of neural data. Overall, the neurally decoded colours qualitatively appear both meaningful and perceptually coherent.

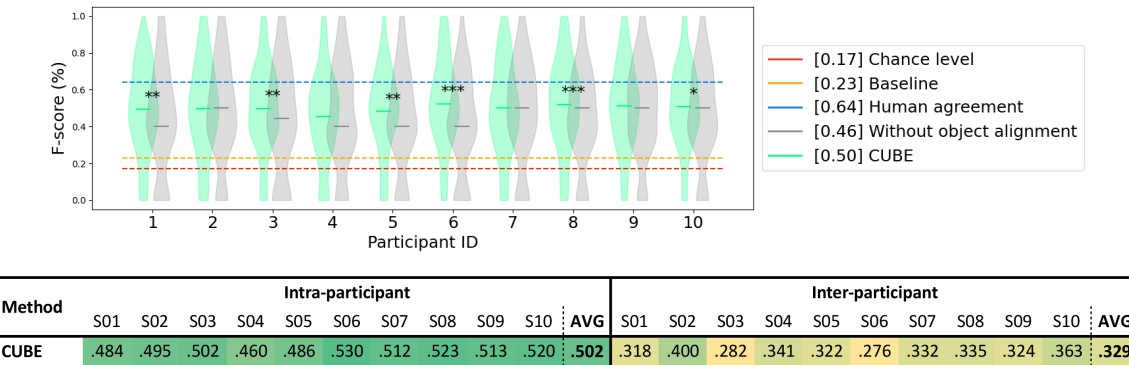

| Method | Intra-participant | | | | | | | | | | | Inter-participant | | | | | | | | | | |
|---|---|---|---|---|---|---|---|---|---|---|---|---|---|---|---|---|---|---|---|---|---|---|
| | S01 | S02 | S03 | S04 | S05 | S06 | S07 | S08 | S09 | S10 | AVG | S01 | S02 | S03 | S04 | S05 | S06 | S07 | S08 | S09 | S10 | AVG |
| CUBE | .484 | .495 | .502 | .460 | .486 | .530 | .512 | .523 | .513 | .520 | .502 | .318 | .400 | .282 | .341 | .322 | .276 | .332 | .335 | .324 | .363 | .329 |

Figure 3: Colour decoding performance of CUBE. Top: F-score distributions over 200 test images. Chance level is shown in red and the noise ceiling from behavioural agreement in blue. Asterisks mark significant differences between models with and without object alignment. Bottom: Average F-scores for within- (intra) and across-participant (inter) training. Table cells are colour-coded from green to yellow as F-scores decline.

## 4 Object decoding

Object decoding in the THINGS EEG and MEG datasets is formulated as a retrieval task. Each network is evaluated in a zero-shot setting, where, among 200 candidate images, the one with the highest cosine similarity to the decoded EEG features is taken as the predicted object category.

Table 1 reports the object decoding accuracy of CUBE and several comparison models on the THINGS EEG dataset (Gifford et al., 2022). CUBE achieves 57% top-1 and 86% top-5 accuracy, representing a 6% improvement over UBP (Wu et al., 2025). This gain is consistent across all participants, indicating robust decoding boost. Accuracy peaks at 66% top-1 and 93% top-5 for the best participant—remarkable given the inherently noisy nature of EEG signals. In inter-participant evaluation, the improvement is more modest—just over 1% in both top-1 and top-5 accuracy—reflecting the substantial challenges of cross-subject decoding, including variability in neural responses (Wei et al., 2021) and individual differences in visual processing (De Haas et al., 2019).

We next examined whether CUBE's improvements generalise across different vision encoders. Figure 4 shows that CUBE yields a statistically significant 5% increase in object decoding accuracy across all seven OpenCLIP encoders (Cherti et al., 2023). To test whether this boost arises solely from the semantic structure of $I^{sam}$ rather than from colour features, we trained CUBE variants that aligned EEG with visual features from greyscale $I^{sam}$, where colour was removed but semantics preserved. These models achieved only a modest 1% average gain, which was inconsistent across encoders and even reduced performance for the CoCa-B32 encoder. Together, these results indicate that colour features provide a substantive and reliable contribution to object decoding.

Table 2 reports object decoding accuracy for CUBE and two comparison models on the THINGS MEG dataset (Hebart et al., 2023). The results closely parallel the EEG findings: CUBE improves intra-participant accuracy by roughly 5% for both top-1 and top-5, and inter-participant accuracy by about 1%. These findings demonstrate that the decoding boost provided by colour features generalises beyond EEG to other neuroimaging modalities.

Table 1: Object decoding performance of CUBE on the THINGS EEG dataset (Gifford et al., 2022) across 200 object categories. Comparison methods from the literature include BraVL (Du et al., 2023), NICE (Song et al., 2024), ATM (Li et al., 2024), IDES (Akbarinia, 2024), VE-SDN (Chen et al., 2024), and UBP (Wu et al., 2025). Table cells are colour-coded from green to yellow as accuracies decrease.

| Method | | | | | Top-1 | | | | | | | | | | | | Top-5 | | | | | |
| --- | --- | --- | --- | --- | --- | --- | --- | --- | --- | --- | --- | --- | --- | --- | --- | --- | --- | --- | --- | --- | --- | --- |
| | S01 | S02 | S03 | S04 | S05 | S06 | S07 | S08 | S09 | S10 | AVG | S01 | S02 | S03 | S04 | S05 | S06 | S07 | S08 | S09 | S10 | AVG |
| Intra-participant | | | | | | | | | | | | | | | | | | | | | | |
| BraVL | .061 | .049 | .056 | .050 | .040 | .060 | .065 | .088 | .043 | .070 | .058 | .179 | .149 | .174 | .151 | .134 | .182 | .204 | .237 | .140 | .197 | .175 |
| NICE | .132 | .135 | .145 | .206 | .101 | .165 | .170 | .229 | .154 | .174 | .161 | .395 | .403 | .427 | .527 | .315 | .440 | .421 | .561 | .416 | .458 | .436 |
| ATM | .256 | .220 | .250 | .314 | .129 | .213 | .305 | .388 | .344 | .291 | .285 | .604 | .545 | .624 | .609 | .430 | .511 | .615 | .720 | .515 | .635 | .604 |
| IDES | .355 | .330 | .305 | .365 | .225 | .345 | .300 | .440 | .355 | .370 | .339 | .645 | .650 | .695 | .715 | .570 | .735 | .700 | .755 | .705 | .705 | .688 |
| VE-SDN | .326 | .344 | .387 | .398 | .294 | .345 | .345 | .493 | .390 | .398 | .372 | .637 | .699 | .735 | .720 | .586 | .688 | .683 | .798 | .696 | .753 | .699 |
| UBP | .412 | .512 | .512 | .511 | .422 | .575 | .490 | .586 | .451 | .615 | .509 | .705 | .809 | .820 | .769 | .728 | .835 | .799 | .858 | .762 | .882 | .797 |
| **CUBE** | .460 | .565 | .615 | .605 | .450 | .595 | .530 | .630 | .555 | .655 | **.566** | .770 | .855 | .895 | .835 | .790 | .880 | .845 | .920 | .855 | .930 | **.858** |
| Inter-participant | | | | | | | | | | | | | | | | | | | | | | |
| BraVL | .023 | .015 | .014 | .017 | .015 | .018 | .021 | .022 | .016 | .023 | .018 | .080 | .063 | .059 | .067 | .056 | .072 | .081 | .076 | .064 | .085 | .070 |
| NICE | .076 | .059 | .060 | .063 | .044 | .056 | .056 | .063 | .057 | .084 | .062 | .228 | .205 | .223 | .207 | .183 | .222 | .197 | .220 | .176 | .283 | .214 |
| ATM | .105 | .071 | .119 | .147 | .070 | .111 | .161 | .150 | .049 | .205 | .118 | .268 | .248 | .338 | .394 | .239 | .358 | .435 | .403 | .227 | .465 | .337 |
| IDES | .090 | .165 | .090 | .135 | .085 | .100 | .075 | .135 | .100 | .190 | .117 | .280 | .340 | .215 | .355 | .260 | .350 | .235 | .315 | .295 | .365 | .320 |
| UBP | .115 | .155 | .098 | .130 | .088 | .117 | .102 | .122 | .155 | .160 | .124 | .297 | .400 | .270 | .323 | .338 | .310 | .238 | .322 | .405 | .435 | .334 |
| **CUBE** | .140 | .175 | .080 | .140 | .135 | .130 | .095 | .125 | .195 | .155 | **.137** | .335 | .410 | .235 | .350 | .310 | .355 | .335 | .385 | .415 | .420 | **.355** |

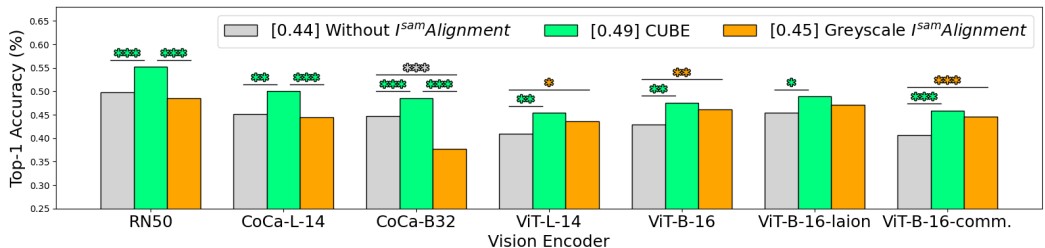

Figure 4: The impact of colour features on object–recognition decoding. Decoding performance for seven vision encoders on the THINGS EEG dataset (Gifford et al., 2022), evaluated across 200 object categories. All encoders were pretrained using OpenCLIP (Cherti et al., 2023). Asterisks denote significant differences between the compared conditions.

Table 2: Object decoding performance of CUBE on the THINGS MEG dataset (Hebart et al., 2023) across 200 object categories. Comparison methods from the literature include BraVL (Du et al., 2023), NICE (Song et al., 2024), and UBP (Wu et al., 2025). Table cells are colour-coded from green to yellow as accuracies decrease.

| Method | | | Top-1 | | | | | Top-5 | | |
| --- | --- | --- | --- | --- | --- | --- | --- | --- | --- | --- |
| | S01 | S02 | S03 | S04 | AVG | S01 | S02 | S03 | S04 | AVG |
| Intra-participant | | | | | | | | | | |
| NICE | .096 | .185 | .142 | .090 | .128 | .278 | .478 | .416 | .266 | .360 |
| UBP | .150 | .460 | .273 | .185 | .267 | .380 | .805 | .590 | .435 | .552 |
| **CUBE** | .165 | .520 | .335 | .235 | **.314** | .435 | .850 | .625 | .480 | **.598** |
| Inter-participant | | | | | | | | | | |
| UBP | .020 | .015 | .027 | .025 | .022 | .057 | .172 | .100 | .080 | .104 |
| **CUBE** | .020 | .050 | .035 | .025 | **.033** | .075 | .175 | .100 | .105 | **.114** |

## 5 Discussion

EEG, whose core technology dates back nearly a century (Berger, 1929), measures tiny fluctuations in ionic potentials to non-invasively record brain activity. The resulting signal is notoriously noisy and has low spatial resolution, reflecting the aggregate activity of billions of neurons (Azevedo et al., 2009; Goriely, 2025). Despite these limitations, EEG has long been an invaluable tool—both clinically and for advancing our understanding of the brain. Recent work suggests that the decoding capabilities of EEG, and neuroimaging more broadly, are undergoing a major leap forward, enabled by AI and large-scale datasets. In particular, EEG benefits from its exceptionally high temporal resolution. We can now decode speech from three seconds of EEG with remarkable accuracy (Défossez et al., 2023), and in the visual domain, emerging work is progressing toward 3D object reconstruction (Guo et al., 2025) and even video decoding (Liu et al., 2024).

Here, we showed that object decoding reaches a remarkable 57% accuracy—far above the 1/200 chance level—from just one second of EEG. Likewise, we demonstrate for the first time that perceived colours in complex natural images can be decoded with high reliability (F-score = 0.5). It is striking that EEG recorded during natural viewing—without any colour cues—can recover colours with reliability approaching that of average behavioural responses. One might expect decoding performance to improve even further (Robinson et al., 2023) if neural and behavioural data were obtained from the same individuals, allowing models to more precisely capture individual differences in perception and colour (Bosten, 2022; De Haas et al., 2019).

### 5.1 The interaction between colour and object

Colour information plays an important role in object recognition (Bramão et al., 2011; Rosenthal et al., 2018), and, conversely, object and scene semantics influence perceived colours (Bloj et al., 1999; Hansen et al., 2006; Akbarinia, 2025). Nevertheless, the interaction between colour and object remains surprisingly little understood (Teichmann et al., 2020; Taylor & Xu, 2021; Gegenfurtner, 2025). Our results from colour and object decoding further support this bidirectional relationship: object alignment boosts colour decoding by 4%, and colour features boost object decoding by 5%.

A central question in visual neuroscience is whether colour is encoded first—with object boundaries emerging later, as in bottom-up region growing—or whether objects are parsed first and colours filled in afterwards, reflecting a more top-down process. To explore this, we compared colour and object decoding within CUBE. A direct comparison, however, is not straightforward. First, evaluation metrics differ: F-score for multi-class colour decoding versus accuracy for single-class object decoding. Second, the ground truths differ: object labels are objective ("this is a snail"), whereas colour annotations represent subjective averages. Third, colour and object are tightly intertwined, creating potential confounds. Colour-diagnostic objects can enhance colour decoding (e.g., bananas are yellow), while natural colour statistics can bias object recognition (Tanaka & Presnell, 1999; Therriault et al., 2009). For instance, a uniformly orange sphere may be decoded as an orange, whereas a striped orange sphere may instead be classified as a volleyball. Fourth, material properties add further complexity (Schmidt et al., 2025), since some materials (e.g., wood) have characteristic colour–texture associations.

Despite methodological challenges, within our framework colour decoding performs relatively better when normalised to chance and noise ceilings: the colour-decoding F-score reaches 0.70 on a 0–1 scale, exceeding the object-decoding accuracy of 0.57. To examine temporal dynamics, we trained models on 100 ms EEG segments from different intervals (e.g., $[0, 100)$, $[100, 200)$, $[900, 1000)$) and also tested shorter windows from stimulus onset up to time $t$. These analyses, shown in Figure 5, reveal that colour decoding becomes statistically significant substantially earlier than

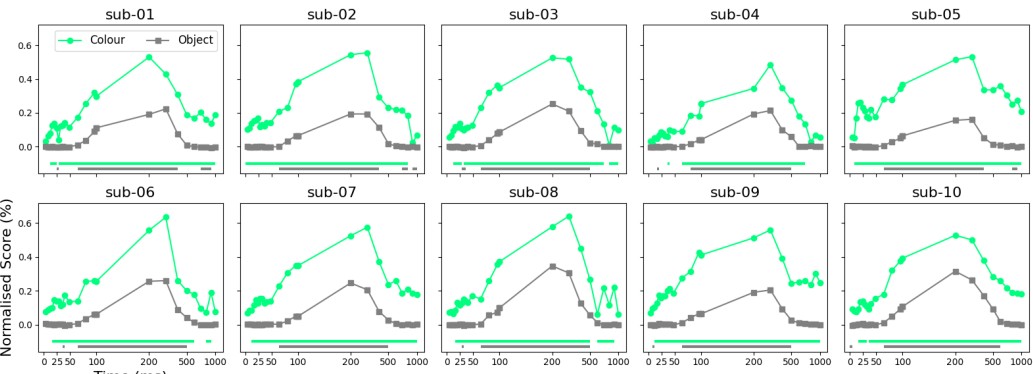

Figure 5: Colour and object decoding across temporal epochs. The x-axis (0 ms) marks stimulus onset; the y-axis shows normalised F-scores (colour) and top-1 accuracy (object), scaled by chance level and noise ceiling. Data points after 100 ms use EEG intervals $[t - 100, t)$, and points before 100 ms from $[0, t)$. Significance lines indicate intervals with decoding above chance ($p < 0.01$).

object decoding—on average at 18 ms for colour versus 41 ms for object. This temporal advantage suggests that colour may play a more prominent role in the earliest stages of visual perception.

## 5.2 Limitations of the EEG decoding

What are the limitations of current EEG decoding frameworks? A major challenge is the substantially lower cross-participant performance compared with within-participant decoding, driven by large individual differences in visual processing across both sensory and perceptual levels (De Haas et al., 2019; Bosten, 2022). Signal quality is further influenced by technical factors such as electrode placement and impedance, which can vary across sessions and participants. One promising direction is to pretrain models on large, diverse EEG datasets (Huang et al., 2025), analogous to large language models, and then fine-tune them for individual participants. This strategy may help bridge the gap between generalisation and personalisation, and could be particularly valuable for practical neuroimaging applications, such as brain–machine interfaces for individuals with severe motor impairments (Chaudhary et al., 2015).

## 6 Conclusion

In this article, we introduced CUBE (ColoUr and oBjEct decoding) and highlighted the importance of jointly representing colour and object features in neuroimaging decoding. Our results show that EEG signals contain reliable, decodable colour information—even during object recognition tasks with no explicit colour cues and under very brief viewing conditions (100 ms). We further demonstrated that incorporating colour features into a standard contrastive-learning alignment framework boosts object decoding by about 5% across participants in both EEG and MEG. Overall, our findings open a novel avenue for future work: theoretically, enabling the investigation of individual colour perception in more ecological settings by decoding colour from neuroimaging signals under naturalistic viewing; and practically, offering potential benefits for applications in brain–computer interfaces and clinical psychology.

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

## A  Qualitative examples

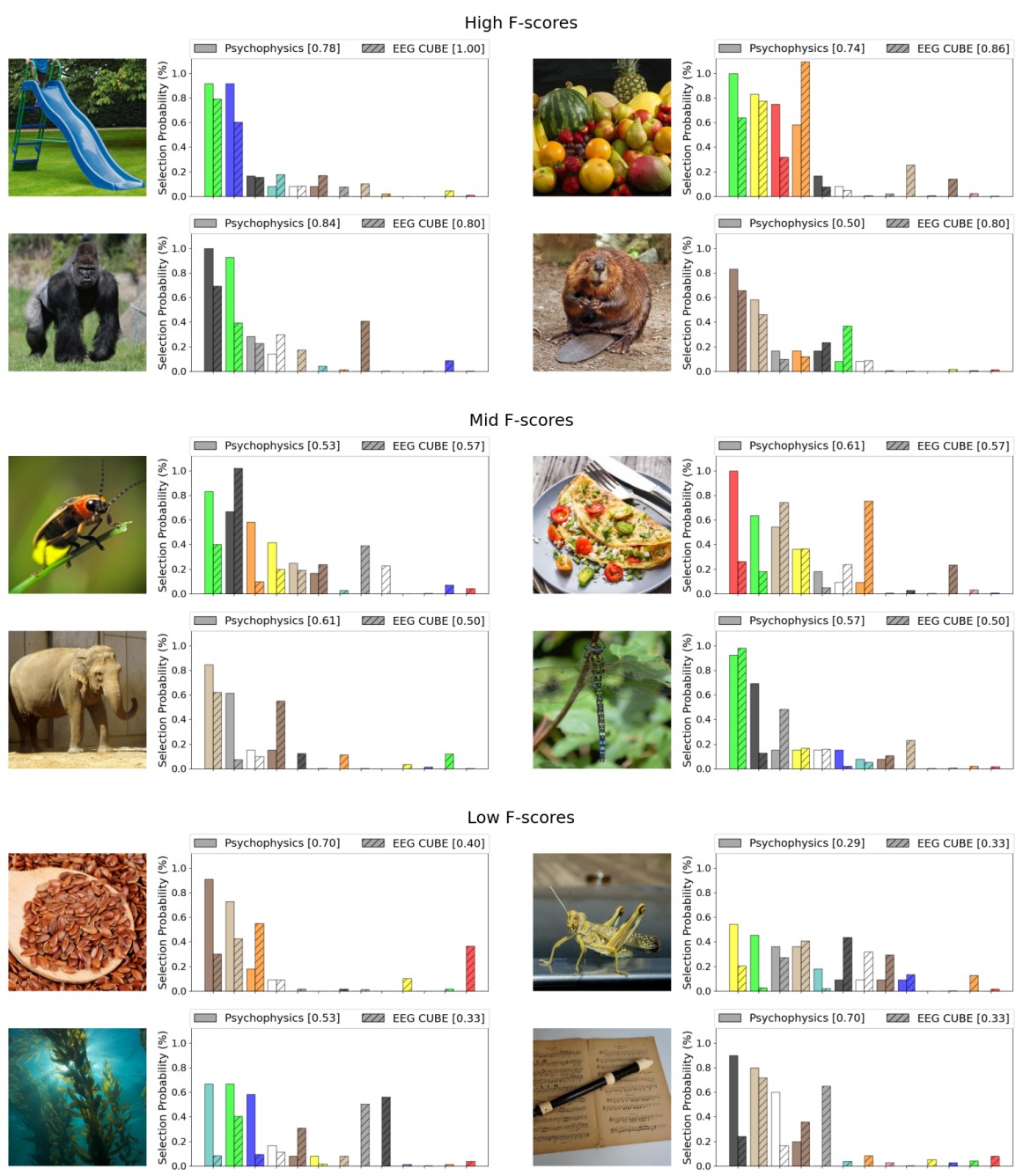

Figure S1: Examples of test images alongside the corresponding colour selections made by human participants in a psychophysical experiment and the colours decoded by CUBE from EEG signals. The reported F-score for the psychophysical data reflects the average inter-participant agreement computed using a leave-one-out strategy, whereas the EEG CUBE F-score represents the agreement between the model's predictions and the average human selections.

