# OpenReview forum: "The hallmark of colour in EEG signals"
_ICLR.cc/2026/Conference — ICLR 2026 Conference Withdrawn Submission_

### Official Review · Reviewer_XxgR · 2025-10-27

**Soundness:** 2
**Presentation:** 3
**Contribution:** 2
**Rating:** 4
**Confidence:** 5

**Summary:**

This paper aligns object and color information with contrastive learning between EEG and vision modalities. Significant performance has been achieved in object and color classification tasks. It also shows that the separate color information enhanced the object decoding.

**Strengths:**

The work focuses on an important task, decoding color information from EEG signals. Hundreds of participants were involved to obtain a psychophysical color ground truth. It showed significant improvement compared to other methods, and the color information significantly improved the overall performance. Several image encoders were compared for demonstration.

**Weaknesses:**

1. There are a few comparisons with other works focusing on color decoding. More comparison would enhance the contribution.
2. It's not clear how the color information was involved in the object decoding. Is that training the contrastive learning method with additional SAM-augmented images?
3. It seems CUBE and UBP achieve largely higher results than previous methods. What is the key increase?
4. The evidence is not convincing to demonstrate that color information was extracted from EEG signals, though the classification results are good. Figure 5 shows that the color has a significant impact on the final performance at a very early stage, where we know the visual processing is not beginning then.
5. It would be beneficial to include more analysis of the psychophysicial experiments to show the consistency between human and machine choice.

**Questions:**

Please see the Weakness part.

---

> ### Author Response · Authors · 2025-11-19
>
> Thank you very much for your constructive questions, suggestions, and comments. Below we provide point-by-point responses to all raised concerns.
>
> ## **Weaknesses**
>
> 1. We have extensively compared CUBE to previous work in both EEG and MEG object-recognition decoding. However, for **colour decoding**, no prior work exists. In this dataset, we created the colour ground truth through large-scale online experiments, and outside the THINGS EEG dataset there are no published networks performing colour decoding from EEG that we can directly compare against. We would greatly appreciate it if the reviewer could point us to any relevant previous work we may have overlooked, and we will gladly incorporate it into the manuscript.
>
> 2. Exactly. The object-decoding pipeline aligns EEG both with CLIP image features derived from the original RGB images and with SAM image features, where each object is represented by a **single colour value**, namely the average of all its constituent pixels. Please see our response to *Reviewer V37C* for details on how we attempted to disentangle the true contribution of colour information to object decoding.
>
> 3. UBP introduced foveation—a well-established mechanism of the visual system—and demonstrated that it substantially boosts decoding performance. We have followed their approach in our contrastive alignment framework.
>
> 4. We are not sure we fully understand this point. We would appreciate clarification from the reviewer so that we can respond more accurately. The EEG decoder receives **only the EEG signal** as input. Colour is decodable earlier than object identity. While this is indeed an interesting observation about how early the visual system interprets a scene, previous work has shown that this can be as early as ~13 ms, as noted in the first paragraph of the Introduction.
>
> 5. A full analysis of the psychophysical experiment on colour perception under short (100 ms) exposures is substantial enough to warrant a separate article, which we are currently preparing. Several factors influence participants’ responses, including context, number of objects, and the complexity of colours. We will gladly mention this in the revised manuscript, but we prefer to present the detailed analysis of the behavioural data separately, as it has independent value beyond the colour decoding in EEG.

---

### Official Review · Reviewer_LSTa · 2025-10-29

**Soundness:** 2
**Presentation:** 4
**Contribution:** 2
**Rating:** 2
**Confidence:** 4

**Summary:**

This paper discusses a method to extract color information about stimuli from EEG recordings of subjects viewing those stimuli. It is evaluated on the Gifford dataset. In a separate data collection effort without EEG, subjects were asked to identify the colors present in the stimuli used in the Gifford dataset. Two major results are presented. In one, an EEG decoder is trained with contrastive learning to produce embeddings aligned with the output of CLIP applied to the corresponding stimuli that have been color enhanced by segmenting the stimuli and colorizing them with the average color per segment. The output of the trained EEG decoder is passed to a color projector that produces a color distribution that is trained with binary cross entropy against the color distributions produced in the second data collection effort. In the second, the standard retieival-based method is used to evaluate the EEG decoder trained as above on color enhanced images against one trained without.

**Strengths:**

The central claim of this paper is that it is possible decode color of stimuli from EEG recordings of subjects viewing the stimuli. This would be important if it is substantiated. But I question this in light of my comments below.

**Weaknesses:**

This whole effort was conducted in the retrieval-based paradigm that is common with the Gifford dataset. Classical decoding work was formulated in either a classification or framework or a detection framework. In these frameworks, test samples are treated independently. In the retrieval-based paradigm, the entire test set is considered at once. For each test EEG sample, the closest test image sample is chosen, after both are mapped to their embeddings. Accuracy is measured by the fraction of test EEG samples for which the correct test image sample is retrieved.

The retrieval-based formulation leads to much higher "accuracies", for example, about 50% on line 294 on a 200-way zero-shot task. But this is misleading. State of the art classification accuracies with the classical classification formulation, where test samples are classified independently, are about 7% for 40-clas on EEG, even when trained with 800 samples per class. This is because the problem formulation is different. In some sense, the retrieval-based formulation is akin to classifying samples with a nearest-neighbor classifier constructed with labeled test set samples.

I would be more convinced of the ability to decode color from EEG recordings if the following simple experiments were conducted and their results reported.

 1. Show subjects screens with solid red/blue/green and perform a classical 3-way classification of EEG recordings, where stimuli are presented for 500ms to 2s. I suspect that you would get about 70% on this 3-way task, where chance is 33% if counterbalanced, because that is about state of the art on 3-way EEG classification from image stimuli presented for that long.
 2. Repeat (1) with 13 colors as done in this paper. I suspect that you would get about 17% (chance is about 7%), because that is about state of the art on 13-way EEG classification.
 3. Repeat (1 and 2) with an RSVP paradigm where stimuli are presented for only 200ms. I suspect accuracy would drop precipitously.
 4. Since you report that subjects report 2 colors per stimulus on average, either do a 13^2=169 way classification on images where the left/top half is one color and the right/bottom half is a different color, or apply a detection paradigm to these stimuli, where you train 13 color detectors, apply all of them to the EEG signal, and evaluate with ROC curves or precision/recall curves. The latter can be applied to more than two colors per stimulus. Do this first with 500ms-2s per stimulus then repeat in an RSVP framework with 200ms per stimulus. I suspect that you will get chance.

If you could get above chance with statistical significance, I would be convinced that it is possible to decode color from EEG. Short of that, I suspect that two things are going on in your results:

 1. Color is confounded with object/scene characteristics, i.e. grass is green, sky is blue.
 2. The whole thing appears to work because of the retrieval-based evaluation paradigm and would fail in a classification or detection paradigm.

**Questions:**

None

---

> ### Author Response · Authors · 2025-11-18
>
> Thank you very much for your constructive questions, suggestions, and comments.
> Below we provide point-by-point responses to all raised concerns.
>
> ---
>
> ## **Weaknesses**
>
> Before addressing the individual points, we would like to emphasise that the **retrieval framework is used only for object recognition**. Since this approach is standard in EEG and MEG decoding, we followed the same framework to enable direct comparison with previous work. We agree that classification accuracy would be lower than retrieval accuracy, but in the THINGS EEG dataset, **classification is not feasible** because there is no overlap between training and test concepts—making the decoding problem even more difficult from another perspective. Nonetheless, this setup is consistent with prior work, so we adopted it for comparability.
>
> We also reiterate that **colour decoding is purely a classification problem**, not retrieval. Each EEG signal is treated independently, consistent with classical neuroimaging decoding practice.
>
> ---
>
> ### **1. Proposed experiments**
>
> The experiment suggested by the reviewer is essentially what we have already performed. Importantly, this is the central point of the article: **colour perception in natural images is far more complex than decoding uniform red/blue/green colour patches**. Participants were allowed to select as many colours as they wished, and we allowed the network to do the same. We report multi-category classification results, which are considerably more challenging and reflective of natural behaviour.
>
> ---
>
> ### **2. Chance level**
>
> The reported **17% chance level** originates from the multi-category classification problem under the assumption that a participant randomly selects two colours on average—matching the behavioural statistics.
>
> The reported **23%** arises from the statistics of the training set and is influenced by two main factors:
>
> 1. **Unequal colour frequency in natural scenes:** Some colours (e.g., beige and green) occur more frequently, leading to a biased colour distribution.
> 2. **Unequal colour frequency in behavioural responses:** Some colours are selected more frequently by participants when reporting perceived image colours.
>
> Naturally, these two factors are related: the human visual system is tuned to the statistics of natural scenes.
>
> ---
>
> ### **3. Decoding colour in other datasets**
>
> We would be delighted to decode colour in additional neuroimaging datasets. We are currently applying the same framework to **THINGS MEG** and **NSD fMRI** datasets.
> However, we did not fully understand the reviewer’s comment regarding the connection to “dropping performance decoding.” We would appreciate clarification so that we can respond appropriately.
>
> ---
>
> ### **4. Number of colours selected by participants**
>
> On average, participants selected **two colours per image**, but the variation is substantial. For example, in the *fruit* image in Figure 6, many more colours were chosen compared with the *gorilla* image shown in the same figure.

---

> > ### Author Response · Authors · 2025-11-18
> >
> > ## **Two Major Questions**
> >
> > ### **1. Evaluation metrics and disentangling colour from semantics**
> >
> > In a multi-class classification problem, simple accuracy is inadequate; therefore, metrics such as **F-measure** or **Jaccard index** are required. These metrics behave similarly, and choosing one over the other does not meaningfully change the results. Irrespective of which metric, a threshold needs to be used to binarise the human colour distributions and model predictions:
> >
> > * While this threshold is indeed arbitrary we chose **1/3**, grounded in the rationale that at least one-third of participants should agree on a colour for it to be regarded as meaningful.
> > * Importantly, varying this threshold does **not** affect the analysis in any meaningful way.
> >   Across all reasonable thresholds, EEG decoding performance remains **substantially above chance and baseline**, in some cases even more so than the values reported in the manuscript.
> > * Additionally, the decoding performance consistently approaches the level of average human agreement from behavioural data.
> >
> > This is the key message of the article: **colour information can be decoded far above chance from EEG recorded while participants viewed natural, complex images**.
> > It is also important to note that even if the “baseline” is treated as chance level—essentially an *informed chance level*, reflecting natural image statistics and behavioural response distributions—the EEG decoding remains far above it, meaning that what is being decoded is genuinely present in the EEG signals.
> >
> > As discussed clearly in the *Discussion*, the THINGS EEG dataset does not allow full disentanglement of colour from semantics. However, removing colour–object associations does **not** reduce decoding to chance—far from it. To summarise:
> >
> > 1. **Images contain multiple colours:** Each image typically contains 3–5 dominant colours. Behavioural responses therefore cannot be derived from the colour of a single object.
> > 2. **Removing object alignment still yields high decoding:** Without object alignment, F-measure remains **0.46**, well above chance. In this condition, the network has **no access to object identity**, so any association must arise purely from EEG signals.
> > 3. **Colour perception is inherently entangled with semantics:** It is natural to think of yellow when seeing a banana. Our results reflect this entanglement, as both colour decoding and object recognition improve when the other is explicitly modelled:
> >
> >    * Object recognition improves by ~5% when colour is modelled.
> >    * Colour decoding improves by ~4% when object alignment is included.
> >
> > ---
> >
> > ### **2. Colour decoding framework**
> >
> > The reported colour-decoding performance is **exclusively** based on a classical multi-category **classification** framework.
> > There is **no retrieval** involved in the colour-decoding analysis.

---

### Official Review · Reviewer_bih2 · 2025-10-30

**Soundness:** 3
**Presentation:** 3
**Contribution:** 3
**Rating:** 6
**Confidence:** 4

**Summary:**

This paper explores the decoding of color information from EEG signals recorded during natural image viewing, and whether color modeling can improve object decoding. The authors create a perceptual color dataset from the THINGS-EEG dataset. They introduce CUBE, a contrastive-learning framework that aligns EEG representations with CLIP features from both original and color-segmented images.

**Strengths:**

- The paper is clearly written and well-organized.
- The construction of a colour perception dataset that complements the THINGS-EEG collection.
- It will likely be useful not only for this study but also for future research on color representation and decoding in the brain.

**Weaknesses:**

- While the idea of integrating color into EEG decoding seems promising, the methodological innovation of the CUBE framework itself seems incremental compared to prior EEG decoding works based on contrastive alignment.
- Does adding color lead to consistent improvements in all feature types or only specific object categories?
- The paper only conducts object recognition task. I am curious about whether color information can help in generation tasks (e.g., reconstructing viewed images from EEG or MEG)?
- Some similar related works are missing. For example, Neuro3D [1] decodes the color information of 3D objects from EEG and also integrates it into the object decoding. Please include the discussion.

[1] Neuro-3D: Towards 3D Visual Decoding from EEG Signals

**Questions:**

Please see weaknesses.

---

> ### Author Response · Authors · 2025-11-18
>
> Thank you very much for your constructive questions, suggestions, and comments. Below we provide point-by-point responses to all raised concerns.
>
> ## **Weaknesses**
>
> 1. While the contrastive learning framework is well established—originating from the OpenAI CLIP paper, where text and images were aligned—and is widely used in neuroimaging decoding since 2023, the **explicit incorporation of colour information for neuroimaging decoding has not been previously explored**. We show that doing so improves object recognition decoding by **5%**.
>
> 2. Colour information improves decoding accuracy across all feature types (Figure 4). However, within the THINGS EEG dataset, we cannot determine whether it improves decoding for **all object categories**, because the test set contains only **one image per object**.
>
> 3. We did not explore image reconstruction in this manuscript, as it lies outside the scope of the current work. All neuroimaging reconstruction algorithms build on contrastive learning frameworks similar to the one used here and then pass features into pretrained image-generation models. Therefore, we believe image reconstruction would also benefit from our proposal, as the input to those models would consist of **better-aligned features**, which we demonstrate in CUBE through higher object-recognition decoding. We will mention this as an interesting direction for future research.
>
> 4. Thank you for pointing this out; we will incorporate discussion of the missing articles.

---

### Official Review · Reviewer_V37C · 2025-10-31

**Soundness:** 2
**Presentation:** 3
**Contribution:** 1
**Rating:** 2
**Confidence:** 5

**Summary:**

The paper introduces CUBE, a color-aware EEG/MEG decoding framework that aligns neural activity with CLIP features from both original images and segmented, color-simplified versions. The authors use THINGS-EEG and collect new 13-color psychophysical annotations, and they report accurate and reliable color decoding from individual seconds of EEG segments. The authors also report a consistent 5% boost in 200-way object recognition when adding the color alignment.They perform a time course analysis which suggests that color becomes decodable before object identity (under 100 ms presentation).

At the core of this work is a scientific claim: EEG signals carry substantial color information during natural vision (also emphasized in the abstract). However, this is not a new idea. Numerous prior studies have already demonstrated that color can be robustly and early decoded from EEG and MEG data. As such, the main finding largely reaffirms what is well established in the literature, using a modernized CLIP-based analysis pipeline rather than uncovering new principles of neural coding or color representation.

**Strengths:**

1. The general question of how color information is represented and decodable from EEG and MEG is interesting and relevant to both neuroscience and machine learning.

2. The paper is well written, clearly structured, and easy to follow. The methods are straightforward, and the figures effectively communicate the main findings.

3. The reported gains in EEG/MEG decoding when including color information are consistent across participants and analyses.

4. The newly collected color annotations for the THINGS-EEG dataset are a useful addition for the community and may support future work on color representation and decoding.

**Weaknesses:**

1. The title claims to identify “the hallmark of color in EEG signals,” but it never becomes clear what that hallmark actually is. The paper shows that color information is decodable and improves object recognition slightly, but that’s not the same as isolating a defining neural signature or mechanism of color processing. There is no clear description of what pattern, feature, or temporal profile qualifies as this “hallmark.” As a result, the title oversells the conceptual contribution. The paper demonstrates that color is decodable (a known result), not that it has uncovered a distinctive neural hallmark of color.

2. This leads to the next imp issue which is the significance and novelty of the report. The claim that EEG signals carry substantial color information is itself not distinctive andhas been known for a very long time. In fact, most studies in neuroscience in fact use black and white images to in fact remove the contribution of color in their decoding analyses. Prior work like from Teichmann, Bartels etc. have repeatedly shown robust color decoding from EEG and MEG signals. The study essentially reaffirms these established findings in neuroscience usinga CLIP-based regression framework and does not offer any new insight into the neural mechanisms of color processing in EEG.

3. The study shows that color helps (perhaps, see below), but it doesn’t explain why or how. There is no analysis disentangling whether the improvement comes from early visual encoding, categorical color associations, or simple low-level luminance differences. It also remains unclear what kind of color information is driving the effect. As such the reported results feels like the first step in a larger set of analyses that should have been carried out but weren’t. It shows that color improves decoding accuracy, but stops there

4. There is an additional concern that the segmentation and uniform color fills may be driving much of the reported effect. These manipulations likely introduce additional shape and part cues that CLIP is already highly sensitive to, making it unclear whether the gains truly reflect color processing in the brain or simply stronger alignment with CLIP’s object–color priors. In other words, the framework may be capitalizing on correlations between object shape and typical color rather than decoding color information per se. Without controls that separate these factors, it’s difficult to attribute the observed improvement specifically to color.

**Questions:**

1. The title claims to identify “the hallmark of color in EEG signals,” but what exactly is meant by this hallmark?

2. Given that the claim that EEG signals carry substantial color information is well established, what is the distinctive significance or novelty of this study? (the difference in timings between this study and Teichman's need to be established on the same dataset)

3. (sorry many questions under this) The study shows that adding color improves decoding accuracy, but what is the underlying reason? Does the effect stem from early visual encoding, categorical color associations, or low-level luminance differences? What kind of color information actually drives the improvement?

4. Could the segmentation and uniform color fills be driving much of the reported gain by introducing additional shape or part cues that CLIP already encodes? How do the authors disentangle true color-related effects from possible object–color priors embedded in CLIP?

---

> ### Author Response · Authors · 2025-11-18
>
> Thank you very much for your constructive questions, suggestions, and comments.
> Below we provide point-by-point responses to all raised concerns.
>
> ---
>
> ## **Weaknesses**
>
> ### **1. Use of the word “hallmark”**
>
> We are very sorry that the word *“hallmark”* has caused confusion. It has been interpreted as too strong. Naturally, we never intended to imply that we have discovered the *true* representation of colour in the brain—if such a unique representation even exists, given how deeply colour perception is entwined with other factors, senses, and semantics.
> We believe, however, that one single word should not be grounds for either favouring or discarding an article.
>
> We originally chose *hallmark* because of its brevity and its fit within a concise title. Our goal was to convey the core message that colour is decodable from natural images, that it interacts meaningfully with object recognition, and that this occurs rapidly. To elaborate, our results show:
>
> 1. Colour is decodable in natural, complex images.
> 2. Colour decoding is possible from neuroimaging data recorded during a colour-agnostic task, essentially during natural behaviour.
> 3. Decoding emerges at very low latencies—essentially as soon as the image is presented for several participants.
> 4. Colour decoding from neuroimaging data approaches behavioural performance.
> 5. Colour decoding consistently boosts object-recognition performance significantly (on average by 5%).
>
> Of course, we have **not** unravelled the neural signature or neuronal mechanisms of colour, nor have we claimed to. Our work offers insights into colour in natural scenes, its temporal dynamics, and its interaction with object perception.
> We are happy to consider a more appropriate title and to clarify this issue in the revised manuscript.
>
> ---
>
> ### **2. Novelty of colour decoding in natural images**
>
> Although colour decoding in neuroimaging has a long history—as noted in our manuscript (second paragraph of page 1)—there is, to our knowledge, **no work showing that colour decoding is possible in natural, complex images**, i.e., images resembling those we encounter in daily life. We would appreciate it if the reviewer could point us to any relevant work we may have missed, which we will happily incorporate into our manuscript to make the discussion more complete.
>
> The closest studies we are aware of are:
>
> 1. **Teichmann et al. (2020)** — a simple foreground object placed on a uniform grey background.
> 2. **Pennock et al. (2023)** — an fMRI study using natural images dataset (NSD), but analysing only saturation and luminance.
>
> Both studies are excellent and advance our understanding of colour perception. We previously discussed them in our manuscript. However, we do not believe our work merely reaffirms their findings. Instead, it **substantially extends** them by demonstrating that colour can be decoded from **natural, complex images**.
> Please also see our response to Weakness #1.
>
> From a technical perspective, colour decoding is **not solely** a consequence of CLIP alignment. Without alignment, decoding reaches **46%**, far above chance. CLIP alignment does indeed boost performance to **50% F-score**, but it is not the sole driver.
>
> ---
>
> ### **3. Scope of the manuscript**
>
> We agree that much remains to be explored regarding colour representation in the human brain, and the reviewer’s suggestions are all very interesting. However, addressing them fully is outside the scope of this already dense manuscript. We have never claimed to provide insights into the hierarchy of visual encoding.
>
> Importantly, decoding multiple colours per image in natural images cannot be explained by a simple low-level luminance strategy. The 13 colours in our palette have different luminance levels, and luminance alone cannot support reliable decoding.
>
> ---
>
> ### **4. Influence of object–colour correlations**
>
> This is an important point, and we discuss it extensively in the manuscript. Colour is inherently multisensory and often entangled with semantic factors. We are currently collecting our own EEG dataset to disentangle effects such as colour diagnosticity, but this is a substantial project on its own.
>
> In the scope of this manuscript within the THINGS EEG dataset, our results speak to both both colour and object decoding and the interaction between them:
>
> 1. Each image contains far more colours than the colour of the main object—typically 3–5 dominant colours. Hence, obtained colour decoding cannot be deduced purely from object identity.
> 2. Removing object alignment reduces decoding only to **F = 0.46**, still far above chance. In this setting, the network has **no access** to object identity (e.g., whether the image contains a gorilla, grass, or sky), so any object–colour association must arise purely from EEG signals.
> 3. Object recognition improves by 5% when colour is explicitly modelled. Symmetrically, colour decoding improves by 4% when object alignment is included.

---

> > ### Author Response · Authors · 2025-11-18
> >
> > ---
> >
> > ## **Questions**
> >
> > ### **1.** Please see our response to Weakness #1.
> >
> > ### **2.** Please see our response to Weakness #2.
> >
> > ### **3. Broader interpretation of results**
> >
> > These are all very interesting questions, and we have not claimed to answer all of them within a single study. Considerably more research is needed. As discussed in the manuscript, natural colour–object associations are the main driving force: when we see an image, we perceive object category and colour jointly.
> > We do not believe luminance is responsible, as our palette spans a wide luminance range across hues.
> > Categorical colour effects may play a role, but the THINGS EEG dataset does not allow us to address this directly. We can comment that n the behavioural data, participants frequently reported feeling more confident about colours of objects they recognised, regardless of whether those objects were colour-diagnostic or man-made.
> >
> > ---
> >
> > ### **4. Contribution of colour vs. segmentation**
> >
> > Object–colour associations are indeed strong (e.g., bananas are typically yellow). To isolate the contribution of colour, we created I^sam from greyscale images. If segmentation alone were driving improvements in object recognition, we would expect improvements here as well.
> >
> > However, alignment with the greyscale I^sam that preserves **semantic segmentation** and removes **colour information** yields only ~1% improvement in object recognition decoding on average, far below the 5% improvement obtained with the coloured version.
> > Additionally, this improvement does not hold across all architectures.
> >
> > Below is a summary of the results:
> >
> > | Model          | Without (I^sam) | With (I^sam-grey) | With (I^sam-colour) (CUBE) |
> > | -------------- | ------------------------ | -------------------------- | ----------------------------------- |
> > | RN50           | 0.50                     | 0.49                       | 0.55                               |
> > | CoCa-L-14      | 0.45                     | 0.45                       | 0.50                                |
> > | CoCa-B32       | 0.45                     | 0.38                       | 0.49                                |
> > | ViT-L-14       | 0.41                     | 0.44                       | 0.45                                |
> > | ViT-B-16       | 0.43                     | 0.46                       | 0.48                                |
> > | ViT-B-16-laion | 0.45                     | 0.47                       | 0.49                                |
> > | ViT-B-16-comm. | 0.41                     | 0.45                       | 0.46                                |
> > | **Average**    | **0.44**                 | **0.45**                 a  | **0.49**                            |
> >
> > We will add this new experiment to **Figure 4** in the revised manuscript, as it allows a clear comparison and highlights the true contribution of colour to object decoding.

---

> > ### Comment · Reviewer_V37C · 2025-11-25
> > **Unaddressed questions**
> >
> > First off, I want to acknowledge the authors for explicitly noting that they haven't identified a hallmark of color decoding (as the title indicates). I hope they do understand how the title may have come across as hyperbolic/overclaiming the central themes. The title definitely needs to change and I would like to know what their intended title would be.
> >
> > Second, the literature on color decoding using is much broader than the 2 studies cited (for example Hajonides et al., 2021 Neuroimaging, Torres-Garcia 2019, among others) and several more in NHP ephys. Notable here that these studies employed experiments directly suited to study the question of color. This is important to clarify the authors' response which is that theirs is a study that uses complex images. But why is it a reasonable choice to use complex images to study color? Naturalistic images contain multiple colours, textures, surfaces, and lighting interactions, making them suboptimal for isolating the specific scientific question the authors are targeting. Researchers typically avoid such stimuli not because colour decoding is impossible, but because complex images introduce confounds that make any neural colour signal difficult to attribute uniquely to colour per se.
> >
> > The response about CLIP alignment does not address this concern. Whether decoding performance is 46% or 50% with CLIP is secondary. The central issue is that decoding in this context is not a clean test of colour representation: with complex images, many scene features covary with colour (semantics, object identity, material properties, lighting), and these covariates are strong enough to drive above-chance decoding even without any explicit colour signal.
> >
> > 3. Thank you for your response but I do not agree that this point is out of scope. My original concern was not about adding an entirely new research question. It is that the scientific claim the manuscript (color decoding from naturalistic images) currently makes cannot be supported without these analyses. The manuscript currently reports the first step in a larger set of analyses that are required to make the scientific conclusion that the paper claims. Those analyses are necessary for the scope the authors themselves have defined. This is part of the reason I feel the paper is premature it its contribution.
> >
> > 4. It is great that the authors are collecting a new EEG dataset to disentangle coufounding factors. I fully agree that such a dataset would be necessary to answer the deeper mechanistic questions. However, this reinforces (rather than resolves) the concern. If colour is inherently multisensory and strongly entangled with semantic information (as the authors state), then the authors implicitly agree that the claims made in the current manuscript are too confounded to support conclusions about colour representation.

---

> > > ### Author Response · Authors · 2025-11-26
> > >
> > > Thank you very much for your response. We do appreciate your time.
> > >
> > > ---
> > >
> > > ## **1. On the use of “hallmark” and title suggestions**
> > >
> > > The Oxford Dictionary defines *hallmark* as *“a distinctive feature”*, which is exactly what we intended with our title: **colour is a distinctive feature in the EEG signal**.
> > > Alternative title suggestions inspired by this notion include:
> > >
> > > * *The footprint of colour information in EEG*
> > > * *Colour featured in EEG responses to natural images*
> > >
> > > We did not intend to exaggerate our findings. As clearly discussed in the *Discussion* and *Limitations* sections, our results show that **colour present in natural images leaves decodable information in the EEG signal**, and our title aimed to reflect that.
> > >
> > > ---
> > >
> > > ## **2. On previous work and the relevance of natural images**
> > >
> > > In the original manuscript, we cited *at least ten* relevant studies that decoded colour information from EEG, including **Hajonides et al. (2021)**. These citations can be found in the Introduction.
> > > We thank the reviewer for pointing out **Torres-García & Molinas**, but once again, that study—like all previous work—investigated only *three* colours and used *simple stimuli*. These works are crucial for understanding colour representation in the brain, yet **none of them address what we investigate here: colour decoding in natural images**.
> > >
> > > While natural images do indeed contain “multiple colours, textures, surfaces, and lighting interactions”, we do not consider this “suboptimal”—we consider it both a *challenge* and an *opportunity*. If the field aims to move toward **real-world EEG decoding**, then interrogating colour in natural images is essential. Colour perception is inherently intertwined with other visual features; thus, understanding colour in ecologically valid conditions requires going beyond simple stimuli.
> > >
> > > From an *application* perspective, natural images are indispensable:
> > >
> > > * In **clinical psychology**, decoding colour from natural scenes would offer far more insight than decoding isolated patches of colour.
> > > * In **BCI/BMI**, users cannot be constrained to predefined sets of stimuli; the goal is precisely to decode colour in unconstrained, real-world scenarios.
> > >
> > > We would very much appreciate clarification on why the reviewer considers the use of natural images a limitation rather than a merit.
> > >
> > > Regarding the comment on removing object alignment: we are unsure why this is viewed as “secondary” in evaluating true colour decoding.
> > > In this condition (F = 0.46):
> > >
> > > * The **only** ground-truth available to the network is behavioural colour data
> > >   (a 13-dimensional vector reflecting all colours perceived in the scene).
> > > * There is **no** object label, object identity, or object alignment information in the learning signal.
> > >
> > > Thus, any decoding performance must necessarily arise from the EEG signal itself.
> > > If the network were to make use of any “non-unique colour cues” (whatever these refer to), such cues would **have to** originate from EEG activity and nowhere else—which is, in fact, what we *aim* to study.
> > >
> > > Our goal is to show that the network can decode *yellow* whether it appears as part of an apple, a banana, a car, or the background grass.

---

> > > > ### Author Response · Authors · 2025-11-26
> > > >
> > > > ## **3. Request for clarification**
> > > >
> > > > We would be grateful if the reviewer could propose a **clear analysis** they believe is missing to demonstrate that colour decoding from naturalistic images is indeed possible.
> > > > We are very willing to add such analyses if the expectation is made explicit.
> > > >
> > > > ---
> > > >
> > > > ## **4. Object recognition and the role of colour**
> > > >
> > > > In our original response to point #4, we provided analyses where colour information was entirely removed from the SAM images. Under this manipulation, **object-recognition accuracy does not consistently improve**, supporting our claim that colour contributes meaningfully to object decoding.
> > > >
> > > > We would appreciate clarification on how the reviewer believes a network **trained solely on colour ground-truth** (with no object labels or semantic information) could achieve **far above chance decoding** if the EEG signal did not contain genuine colour information.
> > > >
> > > > To summarise the logic:
> > > >
> > > > 1. **Participants viewed natural images.**
> > > > 2. **The network receives only EEG as input**—no object category, no semantic label, no image metadata.
> > > > 3. **The network predicts colours in a test set containing entirely new object categories not present during training.**
> > > >
> > > > Given these constraints, we find no viable route for the network to “cheat” or exploit confounds.
> > > > Therefore, we cannot agree with the claim that the conclusions are “too confounded”.

---

### Author Response · Authors · 2025-12-03
**Revised manuscript**

Dear new Area Chair,

We have uploaded the revised manuscript in response to the reviewers’ questions and suggestions. Below, we highlight the key changes and briefly summarise our responses to all four reviewers. More detailed explanations can be found in the point-by-point responses.

1. **Title change.**
   We have updated the manuscript title to **“The footprint of colour in EEG signal”**, avoiding any unintended claims associated with the term *hallmark*, as suggested by reviewer **V37C**.

2. **Novelty and ecological validity.**
   We emphasise that no previous work in the literature has attempted to decode colour under ecological conditions. Our article takes an initial step in this direction by demonstrating the feasibility of colour decoding during naturalistic viewing of complex real-world scenes.

3. **Interaction between colour and object information.**
   In the revised manuscript, we offer a more detailed discussion of how colour and object information interact. Specifically, object alignment improves colour decoding by 4%, and colour features improve object decoding by 5%. We also elaborate further on the temporal decoding profiles of each, addressing comments by reviewer **XxgR**.

4. **Confound concerns regarding object alignment.**
   Although incorporating object alignment enhances colour decoding, we had previously conducted an experiment in which the only training objective was colour classification. In this condition, the network had no access to additional information (see Figure 3, data points *“Without object alignment”*). Therefore, the concerns raised by reviewers **LSTa** and **V37C** about semantic or object information confounds do not apply.

5. **Object decoding and the role of semantic segmentation.**
   For object decoding, we have added results from new experiments conducted during the revision process to clarify the influence of semantic segmentation versus true colour contribution. Please see Figure 4, where the orange bars correspond to *“Grayscale I-sam Alignment”*. These results show that the 5% improvement reported for the proposed CUBE method is largely driven by the colour information available in the I-sam images, and not semantic segmentation information. This addresses the key concern raised by reviewer **V37C**.

6. **Clarification regarding colour decoding methodology.**
   We explicitly state in the revised manuscript that the colour decoding performed here is a classical neuroimaging classification task, not a retrieval task, as was misunderstood by reviewer **LSTa**.

7. **Inclusion of additional related work.**
   We have added and discussed the relevant articles suggested by reviewers **V37C** and **bih2**.

Overall, we believe that the revised manuscript thoroughly addresses all concerns raised by the four reviewers.

---

### Note · Authors · 2026-02-11

I have read and agree with the venue's withdrawal policy on behalf of myself and my co-authors.

---

### Meta-Review · Area_Chair_WgsK · 2025-12-06

**Summary:**

This paper introduces a color-aware contrastive framework aligning EEG features with CLIP features from original and color-segmented images aiming to investigates whether colour information is decodable from EEG.

**Reviewer Concerns:**

While all reviewers acknowledge the presentation quality which is well written, clearly structured, reviewers converge on concerns over significance and novelty of scientific contribution. Decoding color from EEG/MEG has been shown repeatedly in prior literature which is also acknowledged in rebuttal that no previous work decoded colour under "ecological conditions". Current work appears to re-establish known findings using a modern CLIP-based pipeline, rather than uncovering new neural mechanisms or “hallmarks of color” as the title claims. Rebuttal also agrees to change the title to be footprint rather than hallmark.

Multiple reviewers also question whether the retrieval-based evaluation can generalize to classification-based decoding while rebuttal clarify colour coding here is a classical neuroimaging classification task.

Since the conceptual contribution does not meet the bar for acceptance in its current form, this paper is recommended for rejection.

**Reviewer Scores:**

Reviewer V37C: no change

Reviewer bih2: no change

Reviewer LSTa:2->3

Reviewer XxgR: No change.

---

### Decision · Program_Chairs · 2026-01-26

Reject